# *Drosophila* Models for Charcot–Marie–Tooth Neuropathy Related to Aminoacyl-tRNA Synthetases

**DOI:** 10.3390/genes12101519

**Published:** 2021-09-27

**Authors:** Laura Morant, Maria-Luise Erfurth, Albena Jordanova

**Affiliations:** 1Molecular Neurogenomics Group, VIB-UAntwerp Center for Molecular Neurology, Faculty of Pharmaceutical, Biomedical and Veterinary Sciences, University of Antwerp, 2610 Antwerpen, Belgium; laura.morant@uantwerpen.vib.be (L.M.); Maria-Luise.Petrovic-Erfurth@uantwerpen.vib.be (M.-L.E.); 2Molecular Medicine Center, Department of Medical Chemistry and Biochemistry, Faculty of Medicine, Medical University-Sofia, 1431 Sofia, Bulgaria

**Keywords:** aminoacyl-tRNA synthetases, *Drosophila melanogaster*, Charcot–Marie–Tooth neuropathy, disease-modeling

## Abstract

Aminoacyl-tRNA synthetases (aaRS) represent the largest cluster of proteins implicated in Charcot–Marie–Tooth neuropathy (CMT), the most common neuromuscular disorder. Dominant mutations in six aaRS cause different axonal CMT subtypes with common clinical characteristics, including progressive distal muscle weakness and wasting, impaired sensory modalities, gait problems and skeletal deformities. These clinical manifestations are caused by “dying back” axonal degeneration of the longest peripheral sensory and motor neurons. Surprisingly, loss of aminoacylation activity is not a prerequisite for CMT to occur, suggesting a gain-of-function disease mechanism. Here, we present the *Drosophila melanogaster* disease models that have been developed to understand the molecular pathway(s) underlying GARS1- and YARS1-associated CMT etiology. Expression of dominant CMT mutations in these aaRSs induced comparable neurodegenerative phenotypes, both in larvae and adult animals. Interestingly, recent data suggests that shared molecular pathways, such as dysregulation of global protein synthesis, might play a role in disease pathology. In addition, it has been demonstrated that the important function of nuclear YARS1 in transcriptional regulation and the binding properties of mutant GARS1 are also conserved and can be studied in *D. melanogaster* in the context of CMT. Taken together, the fly has emerged as a faithful companion model for cellular and molecular studies of aaRS-CMT that also enables in vivo investigation of candidate CMT drugs.

## 1. Charcot–Marie–Tooth Neuropathy: The Most Common Genetic Affliction of the Peripheral Nervous System

Peripheral nervous system is the anatomical part of the nervous system connecting brain and spinal cord to the other organs in the body, innervating muscles as well as providing sensory input [1]. Hereditary neuropathies is an umbrella term for progressive inherited neurodegenerative disorders involving primarily the peripheral nervous system. Depending on the neuronal population predominantly affected, they are divided into hereditary motor and sensory (also known as Charcot–Marie–Tooth disease, CMT), sensory and autonomic, and motor (also known as distal hereditary motor neuropathies, dHMN) neuropathies. The name CMT is a tribute to the three neurologists (Dr. Jean-Martin Charcot, Dr. Pierre Marie and Dr. Howard Henry Tooth) who independently made the first description of its cardinal signs in the late 1800s [2,3]. The most commonly used estimate of CMT prevalence is 1/2500 individuals, established in a district of Western Norway and ranking it the most common inherited peripheral neuropathy and the most common neuromuscular disorder [4]. Recent meta-analysis of screenings in various populations showed that in fact, the frequency of CMT varies considerably and ranges between 9.7 (Serbian population) and 82.3 (East-Norwegian population) per 100,000 individuals [5,6].

The CMT clinical phenotype is defined by demyelination and/or length-dependent axonal degeneration of the motor and sensory peripheral nerves. The disease onset is usually in early adulthood; however, it varies considerably ranging from first years of life to the fifth or six decade [7]. The disease process starts at the tips of the axons and spreads in a “dying-back” fashion towards the cell bodies. The longest nerves are affected first and are more severely impaired over time. The advancing neurodegeneration causes progressive weakness and wasting of the distal limb muscles leading to motor impairment, sensory loss, and skeletal deformities of the hands and feet [8]. CMT is a slowly progressing disease that usually does not affect life expectancy. There can be a large spectrum of symptomatic severity between individual patients, even between patients within the same family (incl. identical twins) carrying the same mutation [9].

Current classification and nomenclature of CMT combines clinical features, inheritance pattern, histopathological and electrophysiological criteria in order to manage the vast clinical and genetic heterogeneity of the disease and to facilitate the patient’s diagnosis [10]. CMT was divided initially into two major types based on electrophysiological criteria by defining patients with nerve conduction velocities (NCVs) in the median motor nerve < 38 m/s as “CMT type 1” and NCVs > 38 m/s as “CMT type 2” [11]. Neuropathological examination uncovered the underlying reason for the reduction in NCVs as the loss of myelinated fibers aligning CMT type 1 (CMT1) with demyelinating type, as well as onion bulb formations as a result of Schwann cell demyelination and remyelination [12]. CMT type 2 (CMT2) is characterized by progressive dying back of the most distal part of the axons associated with clusters of regenerating nerve fibers [13]. Because the process does not impact myelination, the nerve conduction velocities remain normal or only slightly reduced [13]. A distinctive feature of CMT type 2 are the prolonged compound muscle action potentials signifying the loss of innervation due to axonal degeneration. The initial dichotomous classification of CMT was expanded to include an intermediate disease type (I-CMT) recognized in families where nerve conduction velocities in individual patients range between 25–45 m/s. These patients show a combination of loss of myelination, progressive dying back of the axons coupled with onion bulb formation and presence of regenerating fibers [14]. CMT1 and CMT2 are the most common disease subtypes representing ~70% and ~20% respectively, while I-CMT is diagnosed in less than 10% of the cases [7,15,16].

CMT is a genetic disease where all types of inheritance are observed, including dominant, recessive, X-linked and mitochondrial [17]. Since the mapping of the first CMT locus in 1982, genetic research enabled the identification of more than 100 CMT-causing genes [10]. To account for the genetic heterogeneity, each of the three CMT types is further separated into subtypes based on the underlying inheritance pattern and genetic cause, and is assigned a letter of the alphabet. Most of the encoded products are ubiquitous and housekeeping proteins responsible for endosomal sorting and vesicle trafficking, mitochondrial dynamics and function, axonal transport, myelin structure and integrity and synaptic transmission, among others [17]. Notably, there is no obvious unifying overarching theme explaining the mechanistic involvement of the various types of proteins in the disease process altogether. Furthermore, it is unclear how essential and ubiquitous proteins could cause a very specific degeneration restricted to the peripheral nerves. The unknown cause–effect relationship hampers the comprehensive understanding of the pathomechanistic basis of CMT. Consequently, there is no curative treatment available for any of the CMT subtypes. The management of patients consists mostly of rehabilitative care and symptomatic treatment [10]. Thus, a better understanding of the pathophysiological consequences of genetic abnormalities remains crucial to develop efficient therapies for the patients.

## 2. Aminoacyl-tRNA Synthetases Causing Peripheral Neuropathies

Aminoacyl-tRNA synthetases (aaRSs) are the most represented protein family associated with CMT to date, with dominant mutations described in six genes (Figure 1, Table 1): glycyl-RS (*GARS1*) [18], tyrosyl-RS (*YARS1*) [19], alanyl-RS (*AARS1*) [20], histidyl-RS (*HARS1*) [21], methionyl-RS (*MARS1*) [22] and tryptophanyl-RS (*WARS1*) [23]. Compound heterozygous mutations in lysyl-RS (*KARS1*) [24] have been linked to CMT, however, it remains controversial whether *KARS1* variants are indeed pathogenic due to the lack of comprehensive genetic and functional evidence. Importantly, mutations in all six aaRSs (aaRS^CMT^) cause (predominantly) axonal forms of CMT (CMT2). 

All aaRSs share a common enzymatic housekeeping function: catalyzing the charging of tRNAs with their cognate amino acids prior to protein biosynthesis [25,26]. This two-step reaction starts with amino acid activation by ATP, forming an aminoacyl-adenylate intermediate bound to the enzyme. Then, the aminoacyl-adenylate is transferred to its cognate tRNA, releasing AMP and a charged tRNA that brings the amino acid residue to the growing polypeptide chain in the ribosome. The ubiquitous and essential function of aaRS renders them indispensable for cell viability [27]. Interestingly, several independent studies demonstrated in vitro and/or in vivo that loss of aminoacylation activity is not a common trait among the CMT-causing mutations in GARS1, YARS1, HARS1 and AARS1, implicating another function involved in the pathology (Figure 1) [18,19,21,28,29,30,31,32,33,34,35,36]. Indeed, during evolution, these enzymes acquired additional structural domains allowing them to perform additional activities. Apart from their aminoacylation function, aaRS have also been detected in the nucleus and in the extracellular space, thereby affecting processes such as angiogenesis, hematopoiesis, cytokine signaling and transcriptional regulation [37,38,39,40].

Even though aaRSs are extensively studied and novel insights into their molecular functions have been published recently, the imperative question on how exactly they cause the CMT disease persists. Both in vitro and in vivo approaches have been employed to provide insights into this appealing biological question. Because aaRSs are evolutionary conserved [26], several groups have attempted to study these enzymes and their mutations in lower organisms, including yeast, zebrafish, worm, mice or fruit flies [21,33,35,41,42,43,44,45]. Among those model systems, *D. melanogaster* stands out as the organism where the highest number of CMT-causing aaRS mutations have been modeled and where important breakthroughs have been made. In this review we summarize the recent developments using the fly models and provide an outlook into the future of aaRS studies.

## 3. Modeling aaRS^CMT^ in *D. melanogaster*: Why and How

*D. melanogaster* is an invertebrate model system with a long tradition, where a wide range of established assays exists to evaluate neuronal function in health and disease. Moreover, important biological processes, such as regulation of gene expression, membrane trafficking, cytoskeleton remodeling, neuronal activity or synaptogenesis are conserved at the cellular and molecular level [46]. The genome of *D. melanogaster* contains eight chromosomes—a pair of sex chromosomes and three pairs of autosomes. It is annotated with 13,968 coding genes [47] and is less redundant compared to humans, facilitating genotype–phenotype correlations. Importantly, 60% of genes are conserved between human and *D. melanogaster* and about 65% of human disease-causing genes have a functional orthologue in the fly [48,49,50]. Furthermore, in 2002, a comparative genome analysis predicted 1714 druggable targets [51] rendering this insect an attractive experimental platform for exploring therapeutic strategies. 

*D. melanogaster* has a short generation cycle; it takes up to 10 days at 25 °C for an embryo to develop into a fertile adult and under optimal rearing conditions flies have a median lifespan of approximately 70 days [52,53]. They generate a large number of offspring, as females lay up to 100 eggs per day for up to 20 days, allowing rapid progression of research [52]. The *D. melanogaster* nervous system comprises of a well-studied high-level connectome. This represents an important asset in the investigation of a neurodegenerative disease, allowing extensive and detailed functional analysis. The central nervous system (CNS) is divided into brain and ventral nerve cord. The peripheral nervous system (PNS) contains nerves and sensory organs detecting environmental stimuli (e.g., temperature, light, taste, smell, pressure or air flow) to allow the appropriate behavioral and motor responses. The axons of peripheral nerves are surrounded by glial cells; however, these cells do not synthetize myelin. Therefore, axonal CMT is the most appropriate subtype to study in *D. melanogaster* [54]. To facilitate research, a vast amount of information on genetic, transcription factor–gene, miRNA–gene and protein–protein interactions are integrated into FlyBase, an online bioinformatics database for *D. melanogaster* genetics and molecular biology [55]. In addition, a plethora of publicly available genetic tools are developed for overexpression or downregulation of almost any *D. melanogaster* gene [56,57]. Binary systems such as UAS-GAL4, GeneSwitch or LexA/LexAop combined with CRIPSR/Cas9, RNAi, or transposon-mediated mutagenesis can be used to conditionally activate or downregulate the expression of a targeted gene [49,56,58,59,60,61]. 

An elaborate library of cell type-specific drivers including some that can be compound-, light- or temperature-activated is used to achieve precise spatiotemporal gene expression [49,57,58,59,62]. The reduced but well understood complexity of the *D. melanogaster* nervous system as compared to higher vertebrates combined with the rich genetic toolbox allows an in-depth assessment of neuronal function and pathological dysfunction in both adult and larval stages.

All human CMT-associated aaRSs are conserved in *D. melanogaster* and they share between 41–67% identity at the protein level with their human orthologues [63]. Moreover, most of the amino acid residues affected by the neuropathy-causing dominant mutations are also conserved (Figure 1). *YARS1*-associated neuropathy has been modeled first in *D. melanogaster*, representing the first invertebrate model for CMT neuropathy in general [33]. Altogether, out of five reported mutations for *YARS1* and 21 for *GARS1*, 3 mutations for each of these enzymes were modeled in *D. melanogaster* (Table 2). The spontaneous *GARS1^P278KY^* mutation (corresponding to *GARS1^P234KY^* in human) reported in mouse to cause peripheral neuropathy was also assessed in flies (Table 2) [42,43,64]. This spontaneous mutation changes proline at residue 234 to lysine and tyrosine without affecting the open reading frame [42]. For each model, the UAS-GAL4 binary expression system [33,37,43,44,64] was used to control protein expression at both spatial and temporal level. To create the flies, full length *D. melanogaster* or human *YARS1* or *GARS1* cDNAs (wild type or mutant) were subcloned into expression vectors to allow random [33,37,43,64] or site directed insertion [44] into the fly’s genome, respectively. In most of the studies, fly lines with comparable transgene expression levels were characterized [33,37,43].

As tRNA synthetases are ubiquitous enzymes, overexpression of human mutant *YARS1* (*hYARS1*) or *D. melanogaster YARS* (*dYARS*) was achieved using ubiquitous drivers (Table 3). This led to age-dependent progressive locomotor deficits. Diminished climbing speed was detected in negative geotaxis assays (NGAs), where the natural tendency of flies to climb a wall after agitation was tested (Figure 2, Table 3) [4]. The climbing disability was accompanied by a reduced capacity for jump or flight [4]. Importantly, pan-neuronal or motor neuron restricted expression of human or *D. melanogaster* mutant *YARS1^CMT^* or *GARS1^CMT^* (Table 3), resulted in similar climbing phenotypes [2,3,14]. At the larval stage, the motor deficits manifested as a significant reduction in the rate of muscle contractions in d*GARS^CMT^* models (Table 3) [64]. These data demonstrated in vivo that *YARS1^CMT^* and *GARS1^CMT^* mutations are intrinsically toxic to neurons and that they are causing locomotor deficit, thereby mimicking an important aspect of the disease manifestation in humans. 

The *D. melanogaster* giant fiber system (GFS) is a neuronal circuit mediating an escape response of the fly upon light-off stimuli [69]. It contains the longest neurons in the adult flies; thus, it represents important aspects of the affected neuronal population in the CMT patients. Restricted expression of *hYARS1^CMT^* or *dGARS^CMT^* in the GFS was assessed at both the electrophysiological and the morphological level (Figure 2, Table 3) [33,43]. Intracellular electrophysiological recordings revealed age-dependent synaptic impairment (significantly increased response latency upon a single stimulation and a compelling decrease in the ability to follow high frequency repetitive stimulation) (Figure 2, Table 3) [33,43]. Morphological analysis revealed thin giant fiber terminals with occasional vesicles or constrictions (Figure 2, Table 3) [2,4]. These characteristics are consistent with the age-dependent “dying-back” neuronal degeneration observed in the CMT patients. Caspase activity was not elevated in the GFS of flies expressing *dGARS^CMT^* transgenes, where electrophysiological and morphological signs of synaptic dysfunction were seen, excluding apoptotic cell death as a potential cause for the observed phenotypes [43].

Because in CMT peripheral nerve degeneration starts primarily at the tip of the axon, the synapse between the neuron and the corresponding innervated muscle—called the neuromuscular junction (NMJ)—is likely to be the first site of lesion. In *D. melanogaster* larvae, the synapses between the motor neurons innervating the striated muscle in the body wall are glutamatergic and contain synaptic boutons (Figure 2) [56]. Restricted expression of *hYARS1^CMT^* or *hGARS1^CMT^* in the nervous system or solely in the glutamatergic neurons induced NMJ defects (Figure 2, Table 3) [37,44]. There was a reduction in the total number of synaptic boutons and in the area occupied by neuronal arborizations in *YARS1^E196K^* larvae [37]. Human *GARS1^CMT^* induced significant NMJ morphological defects in a proximo-distal gradient manner as distal muscles were less innervated compared to proximal muscles [44]. Moreover, NMJ analysis of early and late third instar larvae overexpressing *hGARS1^CMT^* revealed progressive muscle denervation over time. Consistently, the larval NMJ defects observed in presence of *dGARS^P234KY^* were translated at the behavioral level in abnormal muscle contractions (Table 3) [64]. These NMJ morphological abnormalities are consistent with the progressive length-dependent axonal degeneration observed in CMT patients, rendering *D. melanogaster* larvae a valuable model to study CMT [18,19].

So far, sensory phenotypes have only been studied in *GARS1^CMT^ D. melanogaster* models at the level of sensory nerve morphology [44]. The larval body wall is innervated by four classes of multidendritic sensory neurons distinct from each other based on their dendritic branching morphology and complexity [72]. When expressed in class IV multidendritic neurons (Table 3), *hGARS1^CMT^* induced a significant reduction in dendritic coverage in the larval body wall (Figure 2, Table 3) [44]. However, the consequence of this phenotype has not been studied at the behavioral level yet. Because these neurons are responsible for thermal nociception [73], mechanical nociception [74] and short wavelength light stimuli [75], their loss might induce behavioral sensory deficits that need to be established in the future. 

Next to the specific features mimicking the CMT pathology, the *D. melanogaster*
*YARS1^CMT^* and *GARS1^CMT^* models presented additional signs of toxicity (Table 3). Ubiquitous expression of human or *D. melanogaster*
*YARS1^CMT^* and *GARS1^CMT^* induced developmental lethality and shortened lifespan in a transgene dose-dependent manner (Figure 2, Table 3) [33,43,44]. Furthermore, GMR-GAL4 driven expression of *hYARS1^E196K^* or *dGARS^P234KY^* in the photoreceptor neurons of the fly eye induced retinal disorganization (“rough eye”) phenotypes in a dose-dependent manner (Figure 2, Table 3) [33,43]. Despite not being a phenotypic characteristic of CMT in humans, the eye disorganization can be used as a sensitive high-throughput read-out for identification of putative CMT-disease modifying genes (see discussion).

Altogether, in several independent studies, all aaRS^CMT^
*D. melanogaster* models presented with common behavioral, electrophysiological and neuropathological phenotypes recapitulating the main hallmarks of disease pathology in humans. Notably, overexpression of the wild type proteins was unremarkable in any of the tested assays. Overexpression of a benign missense variant in *YARS1* (*dYARS^K265N^*) did not cause any phenotypes either. [66]. This indicates that the molecular pathways causing aaRS-associated neurodegeneration are evolutionary conserved and that the *D. melanogaster* models can be used to gain mechanistic insights into the molecular pathogenesis and neurotoxicity caused by the aaRS^CMT^ mutations.

## 4. Mechanistic Insights Derived from *D. melanogaster* aaRS^CMT^ Models

### 4.1. CMT Mutants Active for Aminoacylation Induce Neurodegeneration In Vivo

Historically, the first important mechanistic question about the aaRS-associated CMT was whether the disease is related to the canonical aminoacylation activity. For *YARS1*, it was established that *YARS1^G41R^* and *YARS1^del153^*^−*156VKQV*^ impair the overall aminoacylation activity of the mutant enzymes significantly in vitro. In contrast, the recombinant YARS1^E196K^ protein was fully active for aminoacylation (Figure 1) [33]. These in vitro data were corroborated in the fruit fly in vivo. Hemizygous *dYARS* flies display normal climbing behavior [33]. RNAi-mediated downregulation of *YARS1* (RNAi-dYARS) restricted to the sensory organ precursor (SOP) cells led to absence of dorsal scutellar (dSc) bristles [33]. Co-expression of *hYARS1^WT^* rescued the bristle phenotype suggesting that human and *D. melanogaster*
*YARS1* are functional homologs [33]. Interestingly, overexpression of *hYARS1^E196K^*—but not *hYARS1^G41R^* or *hYARS1^del153^*^−156VKQV^ alleles—fully rescued the bristle phenotype in RNAi-dYARS animals too [33]. This confirmed also in vivo that hYARS1^E196K^ is fully active for aminoacylation. Importantly, this mutant protein still induces the hallmark CMT phenotypes in a manner similar or even stronger than the enzymatically dead versions. 

Similar conclusions were made for *GARS1* using genetic complementation experiments performed in dorsolateral glomerulus 1 (DL1) projection neurons [65]. The highly stereotypical arborization pattern of these olfactory neurons is very sensitive to any defects in their development and maintenance [76,77,78]. In *dGARS*^−/−^ DL1 neurons, dendritic terminals were completely missing, while axons showed pathfinding defects. In addition, the terminal arborizations of the remaining correctly targeted DL1 axons displayed reduced branching [65]. The *dGARS^−/^*^−^ phenotypes were fully rescued by co-expression of the hGARS1^WT^ orthologue [65]. The rescue effect was somewhat reduced by expression of the enzymatically active hGARS1^E71G^ mutant and was completely absent upon expression of the catalytically dead hGARS1^L129P^ enzyme [65]. 

Because both YARS1 and GARS1 holoenzymes act as dimers, it was also tested in vivo if the phenotypes could originate from a dominant-negative effect of the mutant over the wild type allele, thereby hampering the aminoacylation activity of the hetero-dimer. To this end, in larvae pan-neuronally expressing the enzymatically active hGARS1^E71G^ mutant in a wild-type *dGARS* background, formation of heterodimers between the endogenous and the transgenic GARS enzymes were detected. At the same time, the aminoacylation levels in protein extracts from these transgenic flies were comparable to control animals (expressing *hGARS1^WT^* in an endogenous *dGARS* background) [44]. Altogether, these data further confirm that loss of aminoacylation function, either through haploinsufficiency or dominant-negative effect, is not required for mutant aaRS to cause CMT. Rather, they suggest an underlying toxic gain-of-function mechanism [33,37,43,44,79].

### 4.2. CMT-Mutations in YARS1 and GARS1 Reduce Global Protein Translation

Even though it has been established early on in vitro and in vivo that not all CMT-related YARS1 and GARS1 mutats display a loss of aminoacylation function [19,33,34,44], it could not be excluded that they affect the translational machinery or translational regulation in a more indirect manner. In vivo studies in *D. melanogaster* have produced important insights in respect to global protein synthesis [65]. 

The first indications regarding a role of GARS1 in the regulation of protein synthesis were presented by Chihara et al., who identified in an unbiased forward genetic screen *dGARS* as a gene critical for the elaboration and stabilization of terminal arborizations of axons and dendrites of DL1 projection neurons [65]. Because of the striking similarity of the *dGARS*^−/−^ phenotype to the phenotype of other proteins involved in cytoplasmic translation it was suggested that the CMT causing *GARS1* mutations might hamper efficient protein synthesis [65]. The role of *GARS1* in protein translation was further dissected by Niehues et al. using elegant strategies for the labeling of newly synthesized proteins in vivo known as FUNCAT (fluorescent noncanonical amino-acid tagging) and BONCAT (bio-orthogonal noncanonical amino-acid tagging) [44]. The method relies on the incorporation of the noncanonical amino-acid azidonorleucine (ANL) by a *dMARS^L262G^* mutant, which is expressed in a cell type specific manner via the UAS-GAL4 system [80]. Click-chemistry was used to add either a fluorescent (FUNCAT) or a biotin-alkalyne (BONCAT) tag to the ANL-labeled proteins, enabling their efficient semiquantitative detection. In this way, Niehues et al. evaluated protein synthesis of larval motor- and sensory-neurons expressing *hGARS1^WT^*, *hGARS1^G240R^*, *hGARS1^G526R^* and *hGARS1^E71G^* [44]. Misexpression of the wild type human protein did not alter protein synthesis in both motor- and class IV multidendritic sensory neurons. However, protein synthesis rates were significantly reduced to 30–50% of the WT rate in class IV multidendritic sensory neurons upon mis-expression of all three tested *GARS1* mutations (Table 3) [44]. In sensory neurons, *hGARS1^G240R^* and *hGARS1^G526R^* caused a reduction in translational rate to only 40% of WT (Table 3) [44]. Markedly reduced global protein synthesis in the presence of ubiquitously expressed mutant GARS1 proteins was also detectable in adult animals [44]. Interestingly, the same study observed a similar effect in analogous experiments upon misexpression of three CMT-causing *YARS1* mutations (Table 3), with reduction rates of protein translation between 50–80% of WT in motor-neurons and 60–80% in class IV multidendritic sensory neurons [44]. 

The reduction in newly synthesized proteins was not due to increased levels of ubiquitinated proteins, induction of the autophagy pathway or a dominant-negative effect of human mutant *GARS1* on *D. melanogaster* dGARS aminoacylation activity [44]. In line with this, co-overexpression of *dGARS^WT^* was not sufficient to rescue the effects of the human CMT-causing mutations on global protein synthesis in larval motor neurons, suggesting once more a gain of toxic function as the underlying mechanism [44]. Strikingly, in an approach similar to Chihara et al., genetic inhibition of global protein synthesis in larval motor- and class IV multidendritic sensory neurons by expressing constitutively active variants of the *D. melanogaster* 4EBP, a binding protein and inhibitor of eukaryotic initiation factor 4E (eIF4E), caused phenotypes similar to the CMT-phenotypes of *YARS1* and *GARS1* [44]. These experiments focused further the attention on the aaRS’ role in translational regulation to achieve the observed global inhibition effect. 

In a recent study, Zuko et al. dissected the molecular mechanism by which *GARS1^CMT^* mutations inhibit global protein translation [68]. They demonstrated that *GARS1^CMT^* interferes with translation elongation, rather than translation initiation, as overexpression of initiation factors (such as eIF4E, constitutively active S6 kinase or poly(A) binding protein) as well as knock-down of translation inhibition factors (such as Protein kinase RNA-like endoplasmic reticulum kinase (Perk) or 4EBP) neither rescued nor enhanced the translational defects observed in *GARS1^G240R^*
*D. melanogaster* model [68]. Rather, the GARS1^CMT^ mutant proteins sequester tRNA^Gly^ leading to ribosome stalling at Gly codons and activation of the integrated stress response (ISR), ultimately shutting down the global protein synthesis [68]. Notably, overexpression of tRNA^Gly^ partially rescued all phenotypes in their *GARS1^CMT^ D. melanogaster* models (e.g translational defects, larval muscle denervation, larval sensory neuron morphology defects, adult motor deficits, developmental lethality and shortened lifespan) in a dose-dependent manner [68]. Importantly, the rescue effect is cognate tRNA-specific, as overexpression of tRNA^Gly^ had no effect on the neurodegenerative phenotypes in *YARS1^CMT^ D. melanogaster* models [68]. It would be interesting to establish in the future if ribosome stalling leading to global translation inhibition is a common feature of the other CMT-associated aaRS and whether this deficit can be alleviated by overexpression of their cognate tRNAs. 

### 4.3. CMT Mutations Do Not Impair the Subcellular Localization of aaRS in D. melanogaster

In neuronal cells, there is not only translational activity within the cell body, but there also exists local protein synthesis in distal neurite compartments. This extra-somatic protein production is critical during neurogenesis, but it is also essential for continued synaptic transmission and maintenance of mature and ageing neurons [81,82]. Despite being aminoacylation active, the (sub)cellular mislocalization of mutant proteins might disrupt the local translational landscape. To this end, restricted overexpression of *dGARS^WT^* and *dGARS^CMT^* in the GFS of adult flies or ubiquitous overexpression of these proteins in larvae demonstrated comparable cytoplasmic distribution [43,64]. Both WT and mutant proteins were localized to the cell body, axonal bundles containing afferent motor and sensory neurons surrounded by glial cells, and to the neuropil region of axons and dendrites [64]. Within larval motor neurons, both wild type and mutant forms of hYARS1 or hGARS1 were homogenously distributed throughout the cell body, the axons and the neuromuscular junction [44]. Within sensory neurons, hYARS1^CMT^ and hGARS1^CMT^ were detected in the cell body, the axons, and the major dendritic branches in the same manner as the WT enzymes [44]. Notably, different studies reported conflicting findings when the same transgenes were expressed pan-neuronally using different neuronal drivers. Grice et al. did not observe an enrichment of the encoded proteins at the larval NMJ [64], while Ermanoska et al. found comparable and ubiquitous expression throughout the entire neurons [43]. These in vivo results also differ from observations made in differentiated mouse neuroblastoma (N2A) cells, where exogenous hYARS1^WT^ concentrated in granular structures in the growth cones of neurites, at branch points and in the most distal parts of projections, a pattern that was comparable to the distribution of endogenous hYARS^WT^ [19]. This “teardrop”-like distribution was disturbed when hYARS1^G41R^ or hYARS1^E196K^ were expressed [19]. Similar observations were made for hGARS^CMT^ overexpressed in differentiated mouse motor-neuron (MN1) and N2A cells, where the distinctive association of GARS1 to granules and the localization within sprouting neurites was disturbed by the CMT mutations. [83,84]. Taken together, these results highlight the fact that the subcellular localization of YARS1 and GARS1 in fly neurons have only been studied in the context of overexpressed proteins. This entails that more subtle differences in protein localization might have been missed which could be picked up with more sensitive assays, for example if CRISPR/Cas9 based genome engineering and detection of proteins at the endogenous expression levels were used to study the localization of the respective aaRS^CMT^ mutations. 

### 4.4. Nuclear YARS1 as a Transcriptional Regulator

Although protein synthesis takes place in the cytoplasm, aaRSs have also been detected in the nuclei of eukaryotic cells [85]. The human YARS1 protein harbors a nuclear localization sequence (NLS) which facilitates translocation upon oxidative stress to the nucleus of HEK293 cells [86]. Here, it physically interacts with the TRIM28/HDAC1 complex, thereby altering the acetylation levels and activity of transcription factors, such as E2F1 [86]. This nuclear import mechanism is regulated by acetylation of the NLS mediated by the p300/CBP associated factor PCAF [87]. In an effort to investigate the role of nuclear hYARS1 in the context of CMT, Bervoets et al. combined in vivo studies in *D. melanogaster* with molecular experiments in mammalian cells [37]. For this purpose, they created, as a complement to the existing *YARS1^CMT^* disease models, transgenic flies enabling the conditional expression of human *YARS1^WT^* or three CMT mutations each with a disrupting mutation in the NLS sequence (ΔNLS) [37]. In this manner, they were able to exclude overexpressed hYARS1 from the nucleus which allowed them to demonstrate that the nuclear fraction of mutant hYARS1 makes an important contribution to the established neurodegenerative phenotypes due to its enhanced affinity for TRIM28 [37]. These observations were complemented by transcriptome studies of aged adult flies expressing the transgenic *hYARS1* constructs pan-neuronally, demonstrating that translocation of hYARS1 to the nucleus triggers a transcriptional response, which is altered in the case of the *YARS1^E196K^* mutation [37]. This altered signature contains a set of 415 co-regulated genes which has been predicted to be regulated by transcription factors with functions that have been linked to neurodevelopment, dendrite morphogenesis and glucose metabolism [37]. One important aspect of the study by Bervoets et al. is the fact that the *D. melanogaster* models were not only used to test the role of nuclear hYARS1 during neurodegeneration, but that they were also instrumental for testing candidate drugs for their potential beneficial effect. By feeding the ageing *YARS1^CMT^* flies with the respective drugs and assessing developmental lethality and locomotion, the authors were able to demonstrate that pharmacological inhibition of hYARS1 nuclear import via the PCAF inhibitor embelin was much more efficient in rescuing neurodegenerative phenotypes than targeting E2F1 acetylation and activity via the respective modulators resveratrol or dexamethasone [37]. 

### 4.5. GARS1 Is Secreted to Accumulate at the Synaptic Membrane and Interacts with the Plexin-Semaphorin Signaling

AaRS are not only found intracellularly, but some of them can also be secreted. A prominent example in the context of CMT is GARS1 secretion, where an altered conformation of the mutant protein facilitates aberrant interactions with neuronal surface receptors, such as Neuropilin1 (Nrp1) [88] and tropomyosin receptor kinase (Trk1) [89]. GARS1 secretion and aberrant interactions have not only been demonstrated in vertebrate cells but were also observed in *D. melanogaster* models of CMT [64], where restricted overexpression of cytoplasmic *D. melanogaster*
*dGARS^CMT^* in larval muscle or mesoderm induced progressive neuromuscular junction denervation. Misexpression of *GARS1^CMT^* in the larval muscle also selectively caused a very specific wiring phenotype at the larval NMJ, leading to aberrant synaptic connections [90]. These effects are mediated by the WHEP interaction-domain of *GARS1^CMT^* because its deletion prevents the accumulation of *GARS1* mutant protein at the larval NMJ and was sufficient to abolish its toxicity in larvae [64], even though it did not affect GARS1 secretion. These observations indicate that dGARS^CMT^ is secreted into the synaptic cleft where it accumulates at the presynaptic membrane due to its altered binding properties. Here, it exerts its toxicity in a non-cell autonomous manner, leading to denervation (Table 3) [64]. The effect of the aberrant interaction is most likely mediated via the synaptic semaphorin2 receptor plexB, because both proteins were found to co-localize at the membrane and plexB expression and signaling levels modified the viability and motor function defects of *GARS1^P234KY^* larvae as well as the erroneous presynaptic GARS1^P234KY^ accumulations at the NMJ in a dose-dependent manner [64,90]. Taken together with the vertebrate data, this leads to the conclusion that the aberrant interactions of secreted mutant GARS1 with synaptic transmembrane receptors represents an important aspect of CMT-related neurotoxicity. 

### 4.6. GARS1-Induced Neurotoxicity Is Rescued by Inhibition of Sirt2 Deacetylase Activity 

Recently, Zhao et al. reported that the altered conformation of GARS1^CMT^ [88] is capable of disturbing important protein–protein associations, such as the interaction of hGARS1^WT^ with the histone deacetylase Sirt2 [67]. Sirt2 is a major regulator of α-tubulin acetylation [91] which regulates microtubule stability and vesicular transport. The maintenance of the correct α-tubulin acetylation status is therefore especially important to the functioning of the long axons of peripheral nerves. Binding of hGARS1^WT^ to Sirt2 inhibited its enzymatic function in NSC-34 motor-neuron-like mouse cells [67]. The neomorphic conformation of hGARS1 induced by CMT mutations [88] disrupted the Sirt2/hGARS1 interaction. As a consequence, hGARS1^CMT^ was not able to inhibit Sirt2 activity, thus leading to hyperacetylated α -tubulin levels [67]. Importantly, these results were validated in vivo utilizing the *hGARS1^CMT^*
*D. melanogaster* model. Inhibition of Sirt2 using genetic (*Sirt2* knockdown) or pharmacological (AGK2—Sirt2 inhibitor) tools, prevented the climbing and neuromuscular junction deficits induced by expression of *GARS1^G526R^* in motor neurons as described by Niehues et al. (see above and Table 3) [44]. Overall, these results suggest that Sirt2 and proteins acetylation level are involved in CMT-associated *GARS1* neuropathy and that not only gain but also loss of important protein–protein interactions should be considered when investigating the consequence of the open conformations observed in aaRS^CMT^.

## 5. Discussion

Since there are six aaRS that have been linked to CMT, a common disease mechanism has been hypothesized repeatedly. The discovery of such shared neurotoxic signaling pathway(s) is highly desirable as its identification might facilitate the development of drugs for a greater number of individuals afflicted with very similar symptoms. The modeling of *YARS1^CMT^* and *GARS1^CMT^* in *Drosophila* greatly facilitated the current understanding of what in the different models has common etiology and what does not. Overall, it is very encouraging that similar phenotypes were described even though several research groups used different strategies and mutations to generate *D. melanogaster* aaRS^CMT^-models [33,37,43,44]. It is noteworthy that these phenotypes mirror important aspects of the way that CMT manifests in patients; namely, the length-dependent axonal degeneration (e.g., GFS phenotypes in adults and the proximo-distal distribution of NMJ morphological defects in larvae) and the age-dependent locomotor deficits of adult flies. They represent important quantifiable measures that might be very useful when evaluating potential CMT-disease pathway genes or candidate drugs in the future. Other phenotypes do not directly mimic patients’ complaints, but they have proven to be a reliable readout of aaRS^CMT^ related neurotoxicity [33,37,43,44,66]. The *D. melanogaster* eye for example, is an external organ non-essential for viability that has a robust identifiable phenotypic defect (the rough eye) in both *hYARS1^E196K^* and *dGARS^P234KY^* expressing flies (see also Figure 2). This screenable phenotype would require only one generation crossing scheme to enable the identification of CMT-specific aaRS genetic interactors in a high-throughput manner. 

Based on the phenotype data obtained with fly models of *YARS^CMT^* and *GARS^CMT^*, it seems to be promising to create fly disease models for the other aaRS^CMT^. Importantly, all six human enzymes are highly homologous with their fly orthologues and most of their CMT-associated variants affect conserved amino acid residues from humans up to *D. melanogaster* (Table 2), justifying further *D. melanogaster* as a suitable model system. When expanding the portfolio of *D. melanogaster* aaRS^CMT^ models one should however consider the lessons learned so far. First and foremost, unified generation of standardized disease models for all CMT-associated aaRSs is required. Indeed, a prerequisite for the identification of a potential common signaling pathway activated by the CMT-causing aaRS mutations is to use a common modeling strategy and perform all experiments in the most homogeneous genetic background possible (same expression vector, same landing site, same read-out assay, etc.). Here, next to the well-established UAS-GAL4 transgenesis strategy, an attractive new avenue is the CRISPR/Cas9-based introduction of the CMT dominant mutations in the fly genome rather than overexpression of the human transgene. Related to this, one should not forget that all mutant-specific phenotypes described so far are dosage-dependent and are revealed on the background of endogenous protein. Taking advantage of the novel CRISPR/Cas9 editing tools in *D. melanogaster*, one could also consider “humanizing” the models and studying the human aaRS in a knock-out background of the fly orthologue. 

Another aspect requiring further attention is an expansion of the phenotyping strategy. Sensory impairment is part of CMT pathophysiology. So far, the sensory phenotypes of third instar larvae at the behavioral level were not assessed in aaRS^CMT^
*D. melanogaster* models. For example, one could try to analyze temperature and the pain perception of larvae via thermotaxis and nociception assays in a manner similar to the experiments performed in a Rab7^CMT^
*D. melanogaster* model [92]. Apart from larvae, data about behavioral and cellular sensory phenotypes in adult flies have not been collected as of today and warrant further studies. Another open question reflects the potential sex differences in phenotype expressivity, as at least in *YARS1*-associated CMT, men display more severe symptoms than women [93,94]. Usually, only females are included initially in fly behavioral analyses, but more emphasis should be put on the differences with male performance as to reflect the disease characteristics in humans. 

Once unified aaRS^CMT^ disease models have been established, they will enable the systematic investigation of important questions moving forward the aaRS^CMT^ field, including the overarching “common pathway” hypothesis. For example, the systematic large-scale identification of additional, and potentially druggable, genetic modifiers of aaRS^CMT^ toxicity would help deciphering the molecular pathways leading to the CMT disease and whether they are shared among the different aaRSs and with other neurodegenerative diseases having similar pathophysiology. The advantage of such a genetic approach has already been demonstrated by the pilot studies of Ermanoska et al. [43]. The sensitized background induced by the low expression of *hYARS1^E196K^* was used to perform a gain-of function screen for genetic enhancers of the rough eye retinal phenotype. Among 614 tested genes, two genetic enhancers of the retinal degeneration were identified [43]. These two modifiers were demonstrated to enhance the *dGARS^P234KY^*-induced rough eye phenotype too, lending support to the hypothesis of a common molecular pathway in aaRS-associated CMT [43]. Co-expression of *hYARS1^WT^* or *dGARS^WT^* with these modifiers did not induce any rough phenotype confirming the CMT relevance of the interactions [43]. One of these modifiers is *corolla* (CG8316) encoding for a nuclear protein phosphatase 1 binding protein involved in centromere clustering and meiotic nuclear division. The other modifier encodes for an orphan protein with unknown molecular function predicted to be involved in transcriptional regulation and cytoskeletal protein binding [43]. Overall, the knowledge gained with this pilot modifier screen suggests a nuclear involvement in both *hYARS1*- and *hGARS1*-induced CMT, a hypothesis already demonstrated to be viable for *hYARS1* [37]. In addition, these data suggest that shared molecular mechanisms might underly *YARS1* and *GARS1* related CMT. 

In addition, genetic modifier screens in *Drosophila* aaRS would enlighten the mechanistic basis behind clinical variability associated with the aaRS^CMT^ mutations. For example, E71G and D500N pathogenic variants in *hGARS1* have been reported to cause within the same family two clinically distinct disorders, axonal motor and sensory polyneuropathy type 2D (CMT2D) or distal hereditary motor neuropathy type VA (dHMN-VA) [18,95]. Furthermore, in at least two large multigenerational families, asymptomatic carriers of *hYARS1^E196K^* or *hGARS1^L129P^* mutations have been reported [18,93]. This suggests that additional genetic factors could play a role in disease etiology and their identification using the power of *D. melanogaster* genetics might offer attractive translational opportunities. 

A special focus should be on the theme of dysregulated global protein synthesis, which has been reported by Niehues et al. in *D. melanogaster* models for *hGARS1^CMT^* as well as *hYARS1^CMT^*. On one hand, it will be of great interest to test if CMT causing mutations in other aaRS cause comparable phenotypes and reduction in protein synthesis. Interestingly, similar observations regarding protein synthesis have been made for *hYARS1^CMT^* and *hHARS1^CMT^* in vertebrate cells [45,96], suggesting that an overarching theme might indeed be in place. On the other hand, it will be important to dissect the exact molecular mechanisms by which aaRS^*CMT*^ affect global protein synthesis and to establish if and how this effect is related to other findings regarding aaRS and CMT. 

Protein translation is primarily regulated at the level of initiation and aaRS have already been implicated in this process by acting as a scaffold for the assembly of initiation components or by binding to target mRNAs [97,98,99,100]. For example, threonyl-tRNA synthetase (*TARS1*) is part of the translation initiation machinery and interacts with eIF4E, an eukaryotic translation initiation factor involved in directing ribosomes to the cap structure of mRNAs [100]. This interaction allows the recruitment of other translation initiation components to form a machinery structurally similar to the eIF4F-mediated translation initiation [100]. In addition, each aaRS has the dual property to bind both their cognate tRNA and mRNA allowing them to coordinate protein translation and gene expression [98]. Intriguingly, Wei et al. have recently suggested a potential cellular stress mechanism linking the transcriptional dysregulation caused by *hYARS1^CMT^* to downregulation of general protein translation in situations of prolonged stress [96]. In another recent study, Zuko et al. described that *GARS1^CMT^* proteins sequester tRNA^Gly^ both in vitro and in brain tissue of *Gars^C201R/+^* mice, thereby affecting the translational elongation [68]. The depletion of general tRNA^Gly^ pool induces ribosome stalling and leads to activation of the integrated stress response (ISR) [68]. In a parallel study using *Gars^CMT^* and *Yars^CMT^* mouse models, Spaulding et al., performed cell type specific transcriptional and translational profiling demonstrating GCN2-dependent ISR activation specifically in α motor neurons and in a subset of sensory neurons [101]. The general control nonderepressible 2 protein (GCN2) is a serine-threonine kinase playing a key role in modulating the ISR as a response to nutrient deprivation and sensing amino acid deficiency through binding to uncharged tRNA [101]. Genetic or pharmacological inhibition of GCN2 blocked the IRS and alleviated the peripheral neuropathy phenotype in different *GARS1^CMT^* mice models [101]. This suggest GCN2 as an effective therapeutic strategy that could potentially be applied to other CMT-associated aaRS. It might therefore be extremely valuable to follow up on this exciting hypothesis using *D. melanogaster* models for other aaRS^CMT^ to test if a similar link exists not only for *GARS1^CMT^* and *YARS^CMT^* in vivo but also to investigate if the commonality might not be caused by another shared molecular property of all aaRS mutations (e.g., engagement in aberrant protein–protein interactions). 

A systematic comparative study using unified *D. melanogaster* models of aaRS mutations might also reveal the unexpected opposite, namely that no common disease mechanism exists. In either way, important discoveries could be made by utilizing such animal models in combination with classical other disease models, like cellular (including iPSC-based) models, in vivo vertebrate models but also in vitro studies or classical yeast models. The advantage of using *D. melanogaster* will lie in the fact that it allows the study of CMT mutations in the context of a complex nervous system which allows the detailed study of neuronal dysfunction and neurodegeneration. At the same time, it is feasible and affordable to establish models for multiple mutations in multiple aaRS, which also can be studied in a reasonable amount of time, due to the established CMT-related assays. This puts *D. melanogaster* in a unique position among the available CMT models, especially because there has also been a growing interest in using flies for high throughput drug screening. Drugs can be added to the food and readily delivered, allowing the identification of potential candidates that can lead to the discovery of effective therapies [37]. A proof of concept of such strategy was demonstrated recently by Bervoets et al. [37]. This opens the gates for larger scale testing of the same or different drug compounds in the existing and potential future models for aaRS^CMT^ and harbors the possibility for the identification of potential common medicines for a larger group of incurable and disabling hereditary neuropathies.

## Figures and Tables

**Figure 1 genes-12-01519-f001:**
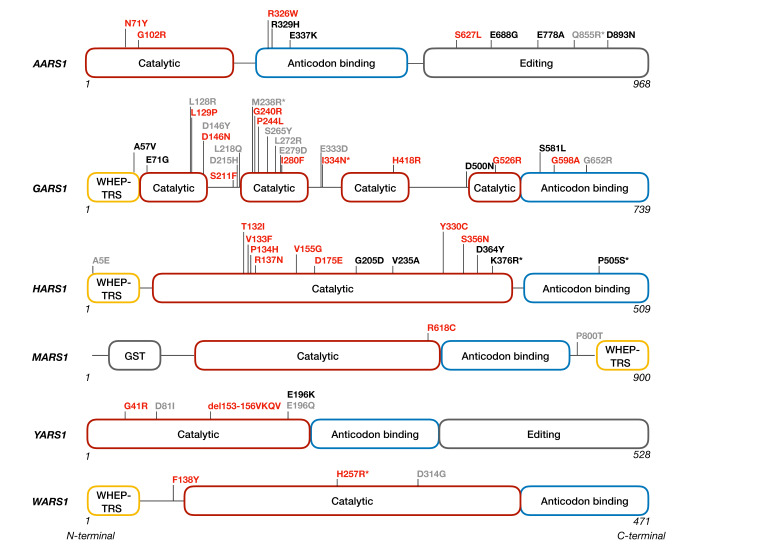
Schematic representation of the location of all CMT-causing dominant mutations reported to date in the domain structure of the corresponding aminoacyl-tRNA synthetases. The CMT mutations can be found in different functional domains. CMT-related mutations affecting the aminoacylation activity are indicated in red, mutations not impairing this activity are depicted in black. The mutations for which the aminoacylation activity remains to be investigated are indicated in grey. The mutations for which the amino acid is not conserved in *D. melanogaster* are labeled with an asterisk (*). The WHEP-TRS domain refers to a highly conserved helix-turn-helix domain of 46 amino-acids found in some of the aaRSs in higher eukaryotes. GST—Glutathione-S-transferase domain.

**Figure 2 genes-12-01519-f002:**
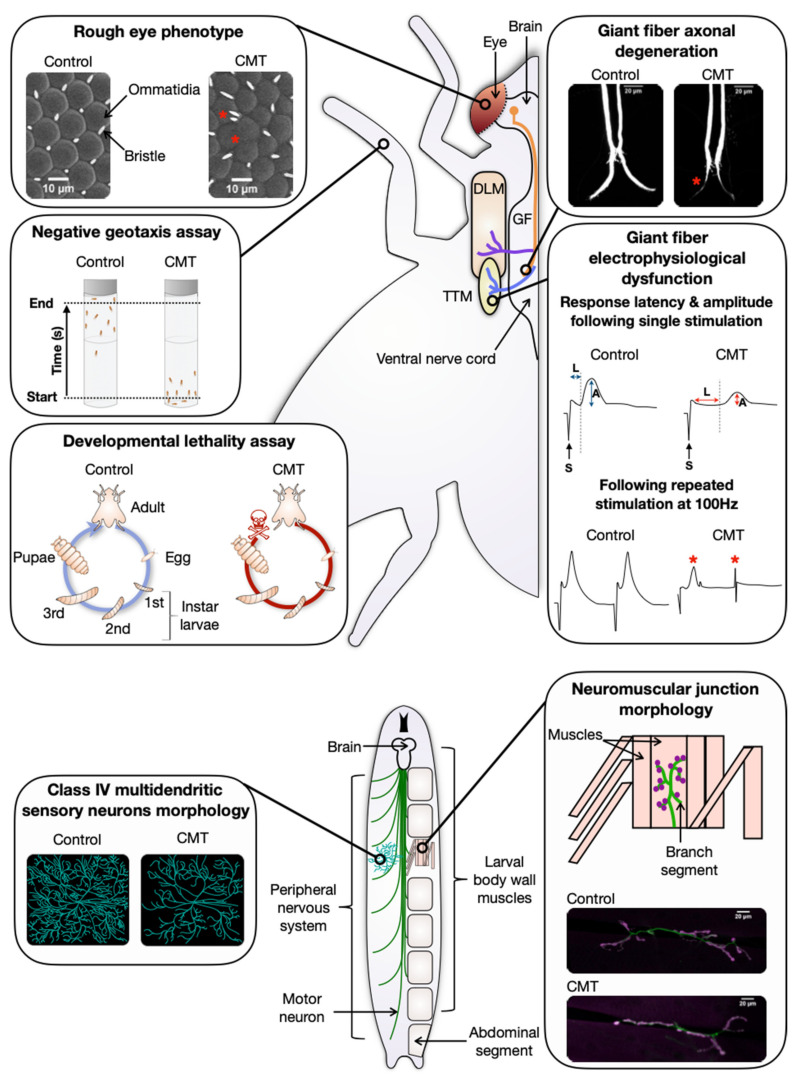
Schematic representation of assays used to evaluate aaRS^CMT^ phenotypes in *D. melanogaster*. Upper panels: assays performed in adult flies. Lower panels: larval assays. The regular and symmetrical organization of the compound eye (red) allows detecting even subtle defects altering the retinal geometry, such as missing bristles and/or merged or misshapen ommatidia (both indicated by red asterisks), resulting in a so-called “rough eye” phenotype [70,71]. The locomotor performance in ageing flies can be evaluated using a negative geotaxis assay (NGA). In this robust behavioral test, the flies are shaken to the bottom of the tube and the climbing speed of the fastest fly to the finish line at 82 mm is measured. The climbing speed is severely decreased in aaRS^CMT^
*D melanogaster* models. The developmental lethality assay assesses the toxicity of mutant proteins at the whole-organism level by counting the number of offspring expressing the transgene that reaches the adult stage and comparing it to the theoretically expected values. Toxic aaRS^CMT^ mutations induce developmental lethality (red skull) ending the life cycle between the first larval stage and the late pupal stage. The central nervous system of *D. melanogaster* is composed of the brain and the ventral nerve cord in which the giant fiber interneuron (GF-orange) contains one of the longest axons. It is part of the giant fiber neuronal circuit, mediating a startle response consisting of jump followed by flight. The morphology of the GF can be visualized by filling with fluorescent dye. The GF innervates the tergotrochanteral jump muscle (TTM—yellow) via the TTM motor neuron (blue) and the dorsal longitudinal flight muscle (DLM—brown) via the DLM motor neuron (purple). The giant fiber in aaRS^CMT^ models is characterized by gross morphological defects, such as abnormally thin axonal terminal (red asterisk) with occasional vesicles or constrictions. GFS electrophysiological dysfunctions in aaRS^CMT^ models include reduction in synaptic strength and reliability, characterized by longer response latency (L) and/or smaller amplitude (A) (red arrows) of the output signal after a single stimulation (S) and a decrease in the ability to follow repetitive stimulations at 100 Hz (red asterisks), respectively. The larval neuromuscular junction (NMJ) morphology is a popular model to evaluate the muscle innervation by motor neurons. Each abdominal segment (grey) in the larval body is composed of unique and identifiable muscles (pink) because of their shape, size, and insertion at the larval cuticle. The NMJ morphological analysis consists of visualizing the motor neurons and their connection to the muscle by using pre-(green) and post-(purple) synaptic markers. The phenotype is assessed by measuring the NMJ length (green) and counting the number of synaptic boutons (purple) and branch segments (green). Class IV multidendritic neurons are sensory neurons with the most complex dendritic arborization, capable of sensing multiple noxious stimuli in *D. melanogaster* larvae. Schematic drawing of class IV multidendritic neurons morphology representing the reduction in dendritic arborization (turquoise) and the decrease in dendritic coverage in the larval body wall due to aaRS^CMT^ mutations.

**Table 1 genes-12-01519-t001:** Aminoacyl tRNA-synthetases causing hereditary neuropathies.

aaRS Gene	OMIM *	Associated Clinical Phenotype	Nomenclature
*AARS1*	613287	Axonal Charcot–Marie–Tooth neuropathy type 2N	CMT2N
*GARS1*	601472	Axonal Charcot–Marie–Tooth neuropathy type 2D Distal hereditary motor neuropathy type VA Distal spinal muscular atrophy type V	CMT2D dHMN-VA dSMA-V
*HARS1*	616625	Axonal Charcot–Marie–Tooth neuropathy type 2W	CMT2W
*MARS1*	616280	Axonal Charcot–Marie–Tooth neuropathy type 2U	CMT2U
*YARS1*	608323	Dominant intermediate Charcot–Marie–Tooth neuropathy type C	DICMTC
*WARS1*	617721	Distal hereditary motor neuropathy type IX	dHMN-IX

*—number in the Online Mendelian Inheritance in Man compendium.

**Table 2 genes-12-01519-t002:** Models for aaRS-related CMT in *D. melanogaster*.

Human Gene	*D. melanogaster* Orthologue % Identity at Protein Level	Modeled CMT Mutation	Reference
*hGARS1*	*GlyRS (dGARS)* 54	E71G	[44,65]
		L219P	[65]
		P234KY *	[43,64]
		G240R	[43,44,64]
		G526R	[44]
*hYARS1*	*TyrRS (dYARS)* 67	G41R	[33,43,44]
		E196K	
		153–156delVKQV	
		K265N	[66]

*—corresponding to P278KY in the mouse orthologue; *dGARS* & *dYARS*: *D*. *melanogaster* orthologues, *hYARS1* & *hGARS1*: human orthologues.

**Table 3 genes-12-01519-t003:** Differential spatial expression of CMT-causing mutations using specific GAL4-drivers results in similar CMT-like phenotypes in both *YARS1^CMT^* and *GARS1^CMT^ D. melanogaster* models.

Phenotype Observed	Transgene	CMT Associated Mutations	Spatial Expression (Driver Used)	References
Age-dependent locomotor deficits	*hYARS1*	G41R, del153-156VKQV, E196K	Ubiquitous (Actin-5c-GAL4)	[33]
			Nervous system (elav-GAL4, nSyb-GAL4)	[33,37]
	*dGARS*	G240R	Nervous system (nSyb-GAL4)	[43]
	*hGARS1*	E71G, G240R, G526R	Glutamatergic neurons (OK371-GAL4)	[44,67,68]
GFS morphological and electrophysiological deficits	*hYARS1*	G41R, del153-156VKQV	Giant fiber (A307-GAL4)	[33]
	*dGARS*	P234KY, G240R		[43]
	*hYARS1*	del153-156VKQV	Pre-synaptic expression in GF (C17-GAL4, R91H05-GAL4)	[37]
	*dGARS*	P234KY	Pre-synaptically TTMn (C42.2-GAL4) or Post-synaptically TTMn (ShakB-GAL4)	[43]
Neuromuscular junction defects *	*dGARS*	P234KY	Ubiquitous (1032-GAL4)	[64]
			Mesoderm (how-GAL4)	
			Muscle (MHC-GAL4)	
	*hYARS1*	E196K	Nervous system (nSyb-GAL4)	[37]
	*hGARS1*	E71G, G240R, G526R	Glutamatergic neurons (OK371-GAL4)	[44,67,68]
Muscle Denervation *	*hGARS1*	G240R	Motor neuron (D42-GAL4)	[44,68]
	*hGARS1*	E71G, G240R, G526R	Glutamatergic neurons (OK371-GAL4)	
Muscle contractions defect *	*dGARS*	P234KY, G240R	Ubiquitous (Tubulin-GAL4)	[64]
			Muscle (MHC-GAL4)	
		P234KY	Mesoderm (how-GAL4)	
			Nervous system (elav-GAL4)	
Reduction of dendritic coverage *	*hGARS1*	E71G, G240R, G526R	Class IV multidendritic sensory neurons (ppk-GAL4)	[44,68]
Rough eye	*dGARS*	P234KY	Eye photoreceptor cell (GMR-GAL4)	[43]
	*hYARS1*	E196K		[33]
Developmental lethality	*hYARS1*	G41R, del153-156VKQV, E196K	Ubiquitous (Actin5c-GAL4, Tubulin-GAL4)	[33]
	*hGARS1*	E71G, G240R, G526R		[44,68]
	*hYARS1*	E196K	Nervous system (nSyb-GAL4)	[37]
	*hGARS1*	E71G, G240R, G526R		[44,68]
	*dGARS*	P234KY		[33,43]
	*dGARS*	P234KY	Nervous system (elav-GAL4)	[64]
			Mesoderm (how-GAL4)	
			Muscle (MHC-GAL4)	
Short lifespan	*hGARS1*	E71G, G240R, G526R	Ubiquitous (Tubulin-GAL4)	[44,67,68]
Inhibition of global protein translation *	*hGARS1*	E71G, G240R, G526R	Class IV multidendritic sensory neurons (ppk-GAL4)	[44,68]
			Ubiquitous (Tubulin-GAL4)	
			Glutamatergic neurons (OK371-GAL4)	
	*hYARS1*	G41R, del153-156VKQV, E196K	Class IV multidendritic sensory neurons (ppk-GAL4)	
			Glutamatergic neurons (OK371-GAL4)	
Transcriptional dysregulation	*hYARS1*	E196K	Nervous system (nSyb-GAL4)	[37]
GARS1^CMT^ accumulation at the synapse *	*dGARS*	P234KY	Ubiquitous (1032-GAL4)	[64]
			Mesoderm (how-GAL4)	
			Muscle (MHC-GAL4)	

* Phenotype observed at larval stage.

## Data Availability

Not applicable.

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
