# Peer review of "Drosophila Models for Charcot–Marie–Tooth Neuropathy Related to Aminoacyl-tRNA Synthetases"

_genes, 2021, doi:10.3390/genes12101519_

Round 1
Reviewer 1 Report
In the review GENES-1357745, Morant and colleagues focus on aminoacyl-tRNA synthetases in Charcot-Marie-Tooth (CMT) neuropathy. This comprehensive requires some conceptual streamlining, but is generally well written and will be received well by the community.
Major point: Is the tile misleading, given that “loss of aminoacylation activity is not a prerequisite for CMT to occur, suggesting a gain-of-function disease mechanism.” l.16? One wonders why aaRS have been grouped together in this review, especially with the emphasis of YARS nuclear, or GARS microtubule-acetylation involvement at the end, before the discussion.
Maybe authors should rephrase and clarify as to: using Drosophila to understand what in the different aaRS models has common etiology, and what does not. For example, the whole discussion around l.383 leads off with a loss-of-function fly GlyRS that seems entirely irrelevant in light of prior strong statements about “underlying toxic gain-of-function mechanism” l.341 for aaRS CMT mutations. Also, to go from toxic GOF to canonical LOF, back to non-canonical GOF in the manuscript makes things seem complicated, hard to follow and unresolved. Maybe better to flow from canonical to non-canonical GOF. Alltogether, there should be more emphasis on whether fly models have helped with the “common pathway hypothesis” l.581. Possibly end on that paragraph, and not on the (re-)distracting l.597 paragraph of loose molecular ends.
Minor points:
the numbering of the sections is off.
l.22 “ex-translational function” is unclear (to the common reader “ex-“ means “former”).
l.23 “the aberrant binding properties of mutant GARS1” sounds as if the reader should know what is being talked about. In addition, one assumes the binding properties refers to the protein, yet the sentence is mentioning the gene (in italics).
l.25 replace “which” with “that” (“which” refers to aaRS-CMT, “that” refers to the fly model).
l.36 “providing the sensory input”; omit “the”.
l.47 replace “may vary” with “varies”
l.59 “divided initially into two types based…” that division based on velocity is somewhat opaque, but maybe more saliently, does it still matter, given the word “initially”? that word sets up an expectation of hearing about “these days, …”. how are CMTs classified these days?
l.86 “causal treatment” sounds odd, since the causes are pointed out as the 100 genes, while the mystery is the link between genes and pathophysiology. maybe just omit “causal”.
l.100 lots of jargony descriptions: CMT2N, CMT2U, dHMN-IX… Many are unexplained, and the ones that are not CMTs are not clinically related/differentiated from ‘classical CMT’. same problem in Fig.1
Figure 1: would it make more sense to have the proteins to scale? is the legend supposed to be flush all the way to the left, like in Fig2? does the legend end on l.115? does MIM refer to OMIM? “The CMT mutations are distributed along the different domains”: they are not really distributed (eg. Yars1), more accurately “can be found in different domains”.
l.155: switch “flies have” and “under optimal rearing conditions”
l.166: “To the benefit of scientific community…”; grammar.
l.173: “On the other hand…” requires a “on one hand…” cognate. furthermore, RNAi, and these days Cas9, is driven by binary systems like Gal4, LexA, thus the statement seems wrong.
l.183, Table1: is identity at the DNA level relevant? or should focus be on amino acid conservation at protein level? (said point made implicitly in l.184!)
l.185: “Historically…” at beginning of a sentence generally refers to a period in the past, and not a single event as stated in this sentence. omit word.
l.187: “five…21…3” is either “five…21…three” or “5…21…3”
l.188: as phrased, the sentence implies that some other mutations were tried, but were not successfully modeled. is that the case? if so, say so, if not, rephrase.
l.189: can you explain P234KY. how is a proline mutated into two different amino acids?
l.194: this is either too much, or too little information. attB may mean nothing to many readers, pUAST is Drosophila jargon.
Table1: the Drosophila genes are called GlyRS and TyrRS. Also, why 4 digits in the % identity (eg 54.77)? is the gene’s DNA >10,000 bp?
Table2 typos: How_GAL4 = how-GAL4 or how24B-GAL4, Elav-GAL4 = elav-GAL4 , Nsyb-GAL4= nSyb-Gal4, Shak-B-GAL4 = ShakB-GAL4; Transgene column should be ahead of mutation column.
l.226: “Importantly, caspase activity was not elevated in the GFS of flies..”, this is only important if we were told beforehand that CMT axon dying back is NOT accompanied by apoptosis. If true, state so.
l.265: buttons = boutons
l.308: “significantly” goes towards the end of the sentence, in front of “in vitro”.
l.374: switch “exists” and “also”
l.377: switch “early on” and “established”
l.412: “analogical” = “analogous”
l.413: “CMT causing” = CMT-causing”, since it’s used as an adjective.
l.464: switch “caused” and “also selectively”
l.522: “landing” = “lending”
l.549: replace “only logic” with “promising”.
l.551: rephrase to “all six human enzymes are highly homologous with their fly…”
l.522: I disagree that homology make Drosophila a suitable model organism, it’s more the face validity of the induced phenotypes. This is especially true in the context of the alleged neomorphic GOF phenotype of the aaRS CMT mutants.
l.574: are there known “gender” differences in CMT, as opposed to “sex” differences? (gender=social, sex=biological)
l.576: should be “compared”
l.579: strike “that”
l.580: the word “moving” requires a direction, presumably “forward”.
l.583: “one hand”..”other” seems misplaced here since the two things discussed are not in opposition, but parallel.
Reviewer 2 Report
This is a very nice and comprehensive review on CMT models in fly. It contains a very good introduction to CMT disease, to aminoacyl tRNA synthetases and to drosophila. Next. Detailed description of the genetic mutations that were introduced in Drosophila melanogaster, assays used to define morphological problems and various interesting experimental findings about impact of AARS in non-canonical mechanism are provided. Finally, the use of Drosophila system is discussed, with advantages and disadvantages being clearly laid out. Altogether, this review is of high interest, includes key details and written in a clear manner. Importantly, it contains nice images to enhance understanding. Of note is the clear description of the advantages of using such a model system to study diseases. Believe that this will be of impact also to other diseases that are studied through the use of a fly model. Overall this is a timely, comprehensive review, on an a subject that is less covered.
I have only few minor comments:
- The sectioning of text is not clear. E.g. there are several ‘sections 2’ (row 90, 141, 304). Section 1.1 (row 305, 342)
- Section 2 (rows 91-104) is a bit confusing. It contains so many names and types of CMT, without any explanation of the differences (e.g. what is the difference between DI-CMT, CMT2D, 2N 2W etc.)
- It will be good to have some kind of description of the different CMT2 variants and their physiological/neuronal manifestation. In both human and fly. This can be in the form of Table, Box or any other representative way. Currently, it is not clear to the reader whether the same manifestation of a mutation is apparent both in human and fly.
- Figure 1 legend is in a format of a regular text.
- (Row 130): an important non-canonical function of aaRS that might be relevant to CMT is binding and regulation of mRNA translation. Recent findings on this topic should be described (e.g. Levi and Arava PLoS Biology 2019 and NAR 2021, Garin S. et al RNA Biology 2021, Jeong, S. J. et al Nature communications 2019, and works from Paul Fox’s lab). These are of importance because CMT might emerge due to problem expression of some mRNAs and not problem with acylation .
